# Characterization of the Far Infrared Properties and Radiative Forcing of Antarctic Ice and Water Clouds Exploiting the Spectrometer-LiDAR Synergy

**Gianluca Di Natale** [1,*] , **Giovanni Bianchini** [1] , **Massimo Del Guasta** [1] , **Marco Ridolfi** [1] , **Tiziano Maestri** [2] , **William Cossich** [2] , **Davide Magurno** [2] **and Luca Palchetti** [1]

1    National Institute of Optics, National Research Council, 50019 Firenze, Italy;
     giovanni.bianchini@ino.cnr.it (G.B.); massimo.delguasta@ino.cnr.it (M.D.G.); marco.ridolfi@ino.cnr.it (M.R.);
     luca.palchetti@ino.cnr.it (L.P.)
2    Department of Physics and Astronomy, University of Bologna, 40126 Bologna, Italy;
     tiziano.maestri@unibo.it (T.M.); william.cossich2@unibo.it (W.C.); davide.magurno2@unibo.it (D.M.)
*    Correspondence: gianluca.dinatale@ino.cnr.it

**Abstract:** Optical and microphysical cloud properties are retrieved from measurements acquired in 2013 and 2014 at the Concordia base station in the Antarctic Plateau. Two sensors are used synergistically: a Fourier transform spectroradiometer named REFIR-PAD (Radiation Explorer in Far Infrared-Prototype for Applications and Developments) and a backscattering-depolarization LiDAR. First, in order to identify the cloudy scenes and assess the cloud thermodynamic phase, the REFIR-PAD spectral radiances are ingested by a machine learning algorithm called Cloud Identification and Classification (CIC). For each of the identified cloudy scenes, the nearest (in time) LiDAR backscattering profile is processed by the Polar Threshold (PT) algorithm that allows derivation of the cloud top and bottom heights. Subsequently, using the CIC and PT results as external constraints, the Simultaneous Atmospheric and Clouds Retrieval (SACR) code is applied to the REFIR-PAD spectral radiances. SACR simultaneously retrieves cloud optical depth and effective dimensions and atmospheric vertical profiles of water vapor and temperature. The analysis determines an average effective diameter of 28 μm with an optical depth of 0.76 for the ice clouds. Water clouds are only detected during the austral Summer, and the retrieved properties provide an average droplet diameter of 9 μm and average optical depth equal to four. The estimated retrieval error is about 1% for the ice crystal/droplet size and 2% for the cloud optical depth. The sensitivity of the retrieved parameters to the assumed crystal shape is also assessed. New parametrizations of the optical depth and the longwave downwelling forcing for Antarctic ice and water clouds, as a function of the ice/liquid water path, are presented. The longwave downwelling flux, computed from the top of the atmosphere to the surface, ranges between 70 and 220 W/m$^2$. The estimated cloud longwave forcing at the surface is (31 ± 7) W/m$^2$ and (29 ± 6) W/m$^2$ for ice clouds and (64 ± 12) and (62 ± 11) W/m$^2$ for water clouds, in 2013 and 2014, respectively. The total average cloud forcing for the two years investigated is (46 ± 9) W/m$^2$.

**Keywords:** cirrus clouds; remote sensing; far-infrared; Antarctic clouds; REFIR-PAD

## 1. Introduction

The importance of clouds in the Earth Radiation Budget (ERB) has been proven by many studies [1,2]. The clouds' contribution is particularly relevant in the Far Infrared (FIR) portion of the spectrum (below 667 cm$^{-1}$), which represents the most energetic band [2] with more than 50% of the outgoing longwave flux. In the polar regions, the FIR contribution can be higher, and it may be as

large as 60% because of the extremely dry conditions and cold temperatures. Nevertheless, there is a lack of spectrally resolved radiance measurements in the FIR, both from satellite and ground-based sites. The scarceness of observations in the FIR was mainly due to limitations in the detectors and optics technology that have been overcome only in recent years [3].

As pointed out in [4], ground-based remote sensing spectral up-looking measurements are very useful to determine the cloud properties relevant to the energy budget. The main reason is that the background is the cold space instead of the surface, as in case of the down-looking measurements. Low sensitivity to cloud properties in passive satellite observations of polar regions is, in fact, due to the similarities of cloud and surface temperature and emissivity. Moreover, ground-based measurements offer the opportunity to test the retrieval algorithms that will be applied for satellite applications. Ground-based measurements are also useful to derive the downwelling flux, which is needed for an accurate characterization of the ERB. As shown in [5], more realistic parametrizations of clouds significantly improve the performance of General Circulation Models (GCM). In particular, the Antarctic climate is very sensitive to ice clouds; thus, changes in their radiative properties might impact climate simulations, not only those limited to the Antarctic continent.

From 2012, the Radiation Explorer in the Far Infrared-Prototype for Applications and Development (REFIR-PAD) Fourier transform spectroradiometer is deployed at the Concordia station, Dome-C, in the middle of the Antarctic Plateau [6]. It is operating continuously with unattended operations to date, providing a wide and unique dataset of downwelling spectral radiances, covering the band from 100 to 1500 cm$^{-1}$ (6.7–100 μm) with a spectral resolution of 0.4 cm$^{-1}$. REFIR-PAD is installed inside an insulated shelter, named the Physics Shelter, about 500 m away from the Concordia base. The instrument points at the zenith through a 1.5 m chimney, to measure a broadband downwelling spectrum about every 15 min. Near-real-time data are accessible from the website http://refir.fi.ino.it/rtDomeC.php. The same Physics Shelter also hosts a backscattering LiDAR. This latter acquires a measurement every 10 min and uses two channels (532 and 1064 nm) to detect the backscattering and depolarization signals originating from up to 7 km above the surface. In addition to these measurements, the Antarctic Meteo-Climatological Observatory installed at Concordia (http://www.climantartide.it/) provides a meteorological database including data of the local automatic weather stations: air temperature, pressure, relative humidity, wind speed, wind direction, and snow temperature at a 5 cm depth. These measurements are archived hourly. Radiosondes measuring temperature and relative humidity profiles are launched daily, at 12 UTC, at a distance of about 500 m from the REFIR-PAD position.

In this work, the synergy of REFIR-PAD and LiDAR is exploited for the retrieval of cloud properties. First, we apply the Cloud Identification and Classification (CIC) algorithm of Maestri et al. (2019) [7] to the REFIR-PAD measured spectral radiance. This algorithm allows discriminating between clear sky and cloudy scenes. For cloudy scenes, the CIC algorithm also determines the cloud thermodynamic phase. For all the detected cloudy cases, the closest acquired LiDAR backscattering profile is processed by the Polar Threshold (PT) [8] algorithm, which determines the Cloud Bottom and Top Heights (CBH and CTH). These results are then used as a priori information by the Simultaneous Atmospheric and Clouds Retrieval (SACR) code [9,10] when applied to the analysis of the REFIR-PAD spectral radiance. SACR simultaneously retrieves temperature and water vapor vertical profiles and cloud parameters, such as the optical depth at visible wavelengths and the ice crystal/water droplet effective diameter.

The paper is structured as follows. Section 2 is dedicated to a detailed description of the algorithms used. Section 3 describes the processed dataset and the retrieval setup. In Section 4, we discuss the results, the errors, and the sensitivity of the retrieved parameters to the assumed ice crystals habit. In Section 5, the longwave downwelling fluxes, as well as ice and water cloud forcings at the surface are assessed. Finally, in Section 6, the summary of the results is drawn.

## 2. Algorithms

### 2.1. CIC: Cloud Identification and Classification

The REFIR-PAD spectra are classified as clear sky, ice cloud, or liquid water cloud by the Cloud Identification and Classification (CIC) algorithm, developed by [7] and here adapted to the experimental conditions that account for up-looking observations. The CIC is a machine learning algorithm based on a Principal Component Analysis (PCA) of the radiance spectra. The CIC algorithm is based on the analysis of the spectral features of the radiance measurements only and does not require any ancillary data or forecast model output for its classification. A limited number of pre-classified spectra, grouped into different classes, are used to train the algorithm. Three groups, called the Training Sets (TSs), are defined for the present work: clear sky, ice cloud, or liquid water cloud. The identification of the TS spectra is performed by visually inspecting the co-located LiDAR backscatter and depolarization profiles. Each TS contains a limited number of spectra (30–50) from the REFIR-PAD database, aiming at describing the multiple conditions of clouds and atmosphere occurring at Concordia. However, due to the large seasonal variability, two separate macro-seasons, thus two groups of three TSs, are considered: a warm one (November-March) and a cold one (April-October). Once the TSs are defined, every other REFIR-PAD spectrum can be ingested by CIC and classified. The CIC algorithm performs a PCA to evaluate how similar the spectrum is to each TS. The similarity between the spectrum and each individual TS is assessed by using a Similarity Index (SI), defined between zero (low similarity) and one (high similarity). The higher the SI, the more the analyzed spectrum has similar spectral features to those contained in the TS (see [7] for more details). A comparison among the SIs results is then performed to establish which TS better describes the spectrum characteristics (the larger SI); the spectrum is thus associated with the proper class. Since three different classes are considered, multiple comparisons need to be accounted for, as shown by the logical diagram in Figure 1. The PCA can be applied to the entire spectral interval covered by REFIR-PAD; however, the interval 380–1000 cm$^{-1}$ is selected for this analysis as the result of an optimization procedure used to maximize the algorithm's performance (Maestri et al., manuscript in preparation).

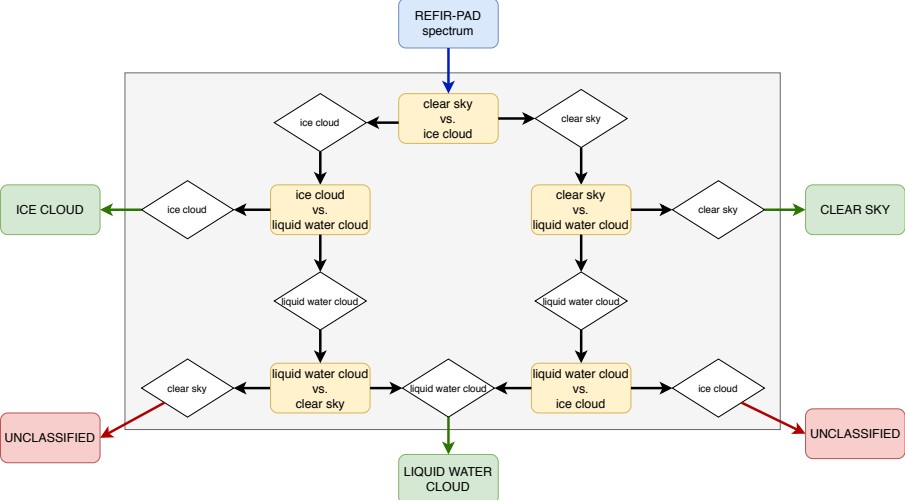

**Figure 1.** Logical diagram of the classification process performed by the Cloud Identification and Classification (CIC) algorithm. The REFIR-PAD spectrum is classified according to the computed Similarity Indices (SIs).

### 2.2. Polar Threshold, Cloud Base, and Top Height Retrieval

When the CIC identifies a REFIR-PAD spectrum as cloudy, the CBH and CTH are derived from the co-located LiDAR backscatter profiles by using a procedure based on the PT algorithm [8]. The PT

code has been developed for the analysis of ceilometer measurements that are performed at a much higher frequency than the LiDAR observations available at the Physics Shelter, at the Concordia station. The PT algorithm detects the CBH from the backscatter profile exceeding a predefined threshold, in combination with noise reduction and averaging procedures. Similarly, the vertical profile of the Signal-to-Noise Ratio (*SNR*) is computed as the ratio between the mean and the standard deviation of three consecutive LiDAR scans, centered on the REFIR-PAD cloudy spectrum timestamp:

$$SNR(z,t) = \frac{\bar{S}(z,t)}{\sqrt{\frac{1}{2M}\sum_{k=-M}^{M}(S(z,t+k)-\bar{S}(z,t))^2}}, \qquad M = 1 \tag{1}$$

where $S(z,t)$ is the LiDAR backscatter signal, at the time $t$ and vertical level $z$, and $\bar{S}(z,t)$ is the temporal mean of the three consecutive LiDAR backscatter signals, computed as:

$$\bar{S}(z,t) = \frac{1}{2M+1}\sum_{k=-M}^{M}S(z,t+k), \qquad M = 1 \tag{2}$$

Since low *SNR* values characterize noisy data, a threshold on the *SNR* profiles is applied in order to determine the cloud position. From the values assessed in [8], an *SNR* threshold of 0.6 is used to discriminate between clear ($SNR < 0.6$) and cloudy ($SNR \geq 0.6$) layers. The CBH is then identified as the first level from the ground, where the *SNR* reaches this threshold. On the other hand, we define the CTH as the level where the *SNR* variation between two consecutive layers reaches the maximum absolute value, due to the sharp variation of the backscattering properties when passing from the cloud top layer to above clear sky.

Figure 2 shows the backscattered signal detected by the tropospheric LiDAR corresponding to the passage of a cirrus cloud (upper panel) over Concordia station. The bottom panel shows the CBH and CTH (magenta and black triangles, respectively) for each REFIR-PAD measurement (dashed back lines) as inferred by the PT algorithm. This algorithm offers a fast solution for the definition of CBH and CTH from the analysis of the LiDAR backscatter signal (averaged every three measurements). Since each average is analyzed independently of the previous and the following one, for some cases, as in the example, sharp variations in the derived cloud geometrical parameters are observed.

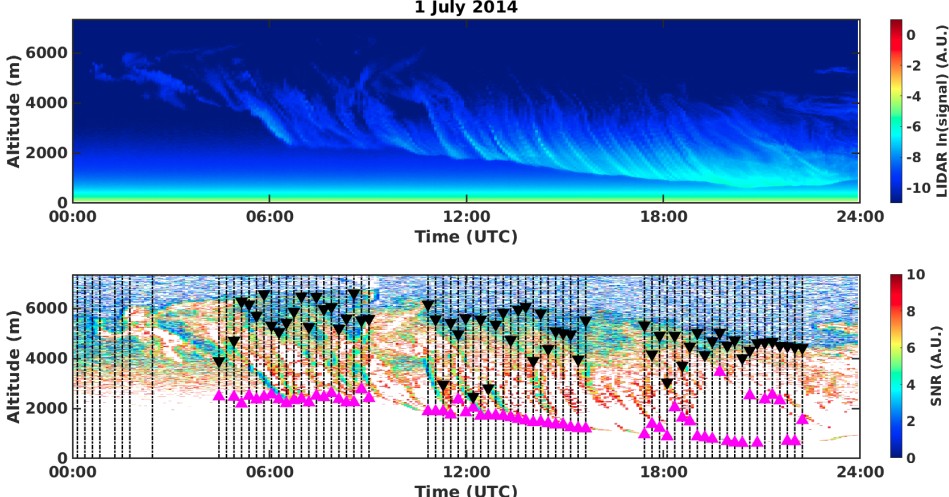

**Figure 2.** Backscattering LiDAR measurements in correspondence to the passage of a cirrus cloud over Concordia. In the **top panel** the backscattered signal as a function of time on the day 1 July 2014. In the **bottom panel**, the PT derived cloud bottom and top heights are marked by magenta and black triangles, respectively. The dashed black lines mark the REFIR-PAD measurements.

## 2.3. Simultaneous Atmospheric and Cloud Retrieval

The SACR code was developed at CNR-INO [9] for the retrieval of atmospheric and cloud properties from REFIR-PAD measurements acquired in the up-looking geometry. The code is made of a validated forward model based on the $\delta$-Eddington approximation to treat the multiple scattering from clouds and aerosols and a retrieval algorithm. The gases' contribution is simulated by using the Line-By-Line Radiative Transfer Model (LBLRTM) [11] code v12.8 with the water vapor continuum MT_CKD aer3.2 [12]. The spectral radiance can be simulated in the whole longwave interval, from far- to mid-infrared, from 100 to 3000 cm$^{-1}$ (3.3–100 µm) in the presence of water, ice, and mixed phase clouds, with also the possibility to mix the contribution of different ice crystal habits. The cloud optical coefficients are taken from specific databases developed by Ping Yang and colleagues [13] for different particle habits. The SACR code can retrieve the effective diameters of ice crystals and water droplets, the ice/water optical depth at the visible wavelengths, and the ice fraction in the case of mixed phase clouds. These cloud parameters are retrieved together with temperature and water vapor vertical profiles. An additional retrieval parameter allows adapting the Instrument Line Shape (ILS) model to the actual ILS. The retrieval is based on the Optimal Estimation (OE) approach [14]. The algorithm finds the solution as the minimizer of the following cost function:

$$\chi^2 = (\mathbf{y} - \mathbf{F}(\mathbf{x}))^T \mathbf{S}_\text{y}^{-1} (\mathbf{y} - \mathbf{F}(\mathbf{x})) + (\mathbf{x} - \mathbf{x}_\text{a})^T \mathbf{S}_\text{a}^{-1} (\mathbf{x} - \mathbf{x}_\text{a}) \tag{3}$$

where $\mathbf{y}$, $\mathbf{x}$, $\mathbf{F}$, and $\mathbf{x}_\text{a}$ represent the vector of the measurements, the state vector, the forward model simulated spectrum, and the vector of the a priori parameters, respectively. $\mathbf{S}_\text{y}$ and $\mathbf{S}_\text{a}$ denote the error Variance-Covariance Matrices (VCM) of the measurements and of the a priori estimate $\mathbf{x}_\text{a}$. The measurement error VCM is built considering both the spectrally uncorrelated error component due to the Noise Equivalent Spectral Radiance (NESR) and the fully correlated component due to the calibration error. The minimum of the cost function (3) is found using the Levenberg–Marquardt method, with the following iterative formula [14]:

$$\mathbf{x}_{i+1} = \mathbf{x}_i + [\mathbf{K}_i^T \mathbf{S}_\text{y}^{-1} \mathbf{K}_i + \lambda_i \mathbf{D}_i + \mathbf{S}_\text{a}^{-1}]^{-1} [\mathbf{K}_i^T \mathbf{S}_\text{y}^{-1} (\mathbf{y} - \mathbf{F}(\mathbf{x}_i)) - \mathbf{S}_\text{a}^{-1} (\mathbf{x}_i - \mathbf{x}_\text{a})] \tag{4}$$

where $\lambda_i$ is the Levenberg–Marquardt damping factor at iteration $i$, $\mathbf{K}_i$ is the Jacobian matrix of the forward model, and $\mathbf{D}_i$ is a diagonal matrix as shown in [9]. When convergence is reached (i.e., when the relative variation of (3) within two subsequent iterations is smaller than a pre-defined threshold), the error on the retrieved parameters can be calculated through the formula [14]:

$$\mathbf{S}_\text{x} = (\mathbf{K}^T \mathbf{S}_\text{y}^{-1} \mathbf{K} + \mathbf{S}_\text{a}^{-1})^{-1}. \tag{5}$$

The REFIR-PAD Instrument Line Shape (ILS) is modeled as a linear combination of a sinc function, which is the main contribution, and a sinc$^2$ function, modeling the self-apodization owing to the finite internal solid angle $\Omega$ of the instrument. For REFIR-PAD, this angle is estimated to be $\Omega = 0.00087$ sr [15]. In the wavenumber domain, the resulting ILS is expressed as [16]:

$$\text{ILS}(\nu) = \alpha_\nu \cdot \text{sinc}(\frac{\nu}{\Delta\nu}) + (1 - \alpha_\nu) \cdot \text{sinc}^2(\frac{\nu}{2\Delta\nu}) \tag{6}$$

where $\Delta\nu$ is the spectral resolution (equal to 0.4 cm$^{-1}$ in our case) and $\nu$ represents the wavenumber. The coefficient $\alpha_\nu$ in (6) takes into account the self-apodization due to the finite solid angle $\Omega$ and is calculated as [15]:

$$\alpha_\nu = \text{sinc}(\frac{L_\text{MAX}\nu\Omega}{2}) \tag{7}$$

where $L_\text{MAX}$ is the maximum optical path difference of the interferometric scan, which is related to $\Delta\nu$ through the relation $\Delta\nu = 1/(2L_\text{MAX})$. Thus, since $\Omega$ is known, in principle, the ILS should be fully

characterized. Despite that, the finite aperture $\Omega$ and the instability of the reference laser contribute to shifting the frequency grid, such that $\nu$ in the expressions (6) and (7) becomes [16]:

$$\nu' = (1 + \beta) \cdot \nu \tag{8}$$

where $\beta$ is the relative frequency shift factor, which is generally of the order of $10^{-6}$ and must be retrieved. According to these considerations, the state vector in (3) and (4) is built as:

$$\mathbf{x} = (D_e, \text{OD}_v, \mathbf{U}, \mathbf{T}, \beta) \tag{9}$$

where $\mathbf{U}$ and $\mathbf{T}$ are vectors representing water vapor and temperature profiles at fixed pressure levels. These profiles are retrieved along with the cloud parameters, $\text{OD}_v$ (the ice/water optical depth at visible wavelengths) and $D_e$ (the effective diameter of ice particles or water droplets). $D_e$ is defined as [17]:

$$D_e = \frac{3}{2} \frac{\int_{L_{\min}}^{L_{\max}} V(L)n(L)dL}{\int_{L_{\min}}^{L_{\max}} A(L)n(L)dL} \tag{10}$$

where $L$ is the maximum length of the ice crystal for a specific habit or the radius in the case of spherical water droplets and $V$ and $A$ represent the volume and the projected area of the particles, respectively.

The optical depth in the FIR band ($\text{OD}_{\text{FIR}}$) is related to the visible optical depth $\text{OD}_v$ by means of the following formula:

$$\text{OD}_{\text{FIR},\nu} = \text{OD}_v \frac{\langle Q_e \rangle_{\text{FIR},\nu}}{\langle Q_e \rangle_v} \tag{11}$$

where $\langle Q_e \rangle_{\text{FIR},\nu}$ is calculated as in [18] and $\langle Q_e \rangle_v$ is the extinction efficiency factor in the visible wavelengths. This factor can be assumed constant and equal to two because of the large size parameters ($\frac{\pi D_e}{\lambda}$), usually greater than 20, for the typical visible wavelengths. Finally, in order to take into account the multiple scattering effect at different heights, the normalized LiDAR backscattering signal $S(z)$ at height $z$ is used to modulate the vertical shape of $\text{OD}_v$ (9) within the cloud, according to:

$$\text{OD}_v(z) = \text{OD}_v \cdot S(z) \tag{12}$$

## 3. Retrieval Setup

### 3.1. Selected Dataset of Spectral Radiances

Spectral radiances are acquired by REFIR-PAD continuously about every 15 min, in the band 100–1500 cm$^{-1}$, with a spectral resolution of 0.4 cm$^{-1}$. The dataset used in this work includes spectra from all seasons of the years 2013 and 2014. The measurements are performed within the Italian PNRA (Programma Nazionale di Ricerce in Antartide, National Research Program in Antarctica), in the scope of the Antarctic projects PRANA (Proprietà Radiative del vapore Acqueo e delle Nubi in Antartide (Radiative Properties of Water Vapor and Clouds in Antarctica)) and CoMPASS (Concordia Multi-Process Atmospheric Studies).

From the selected dataset, two-thousand five-hundred twenty-three measured spectra are classified as cloudy by the CIC algorithm. These include spectra affected by ice, water, and mixed phase clouds. However, only for 1439 spectra, the SACR retrieval converges to a minimum of the cost function ($\chi^2$) smaller than three. Among them, one-hundred twenty-nine spectra are acquired in autumn, 77 in spring, 541 in summer, and 692 in winter.

In this sub-dataset, ice clouds occur in all seasons of the two years considered, while water clouds are detected only in summer time, when insolation is stronger. In Figure 3 are shown some examples of measured spectral radiances in the presence of ice (upper panel) and water (lower panel) clouds, with different visible optical depths: 0.24 and 0.64 for ice clouds and 0.74 and 2.31 for water clouds.

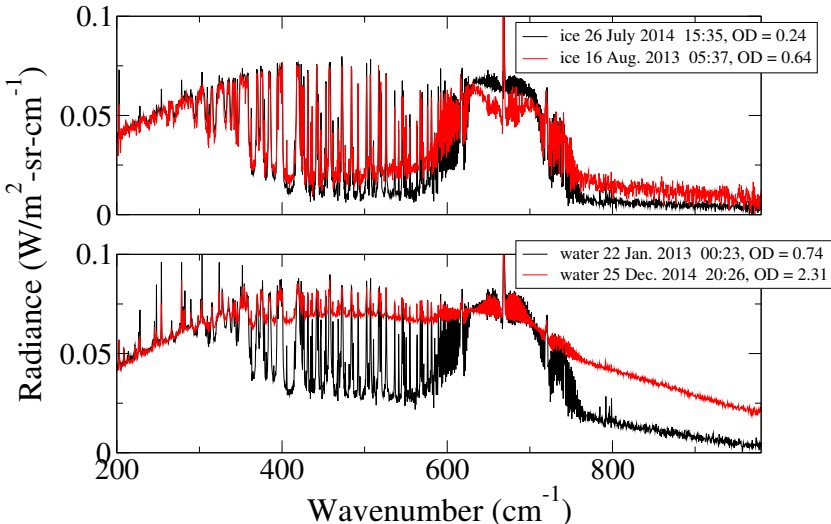

**Figure 3.** Examples of measured spectral radiances in the presence of ice (**top panel**) and water (**bottom panel**) clouds. Measurement date/time and retrieved values of $OD_v$ are shown in the plot's key.

### 3.2. Operational Choices

For the retrieval of cloud and atmospheric parameters with the SACR algorithm, only the spectral interval between 200 and 980 $cm^{-1}$ is used. This portion of the spectrum contains sufficient information on the retrieval parameters and is measured with a good signal-to-noise ratio. Depending on the cloud phase determined by the CIC, the SACR algorithm automatically points to the correct auxiliary database to be used and sets up the state vector to determine ice or water cloud properties.

Since the particle habit distribution is not known, the column-like ice crystals are used in the cloud model. This habit is commonly observed in precipitation in Antarctica [19,20]. In particular, the mentioned papers show that below $-40\,°C$ and with low supersaturation, the ice crystals grow as solid prisms (columns) with the ratio $c/a$ usually up to about five or six. The ratio $c/a$ is greater than one for columns and is computed using the length of the $c$-axis to $a$-axis in the apparent shape. "Long prisms", i.e., columns with $c/a \geq 10$, grow in the narrow range between $-45$ and $-50\,°C$. On the other hand, Lawson et al. in [21] showed that other types of habits are found in Antarctica, such as plates, rosettes, and irregular and complex aggregates. Then, in Section 4.4, a sensitivity study of the retrieved parameters by varying the ice crystal habit is performed.

The ice crystals' optical coefficients, such as the extinction and absorption efficiencies, the single scattering albedo, and the asymmetry factor, were obtained from [13]. The coefficients are tabulated as a function of the wavenumber $v$ and the maximum length $L$. For liquid water droplets, the Mie theory is used to derive the optical coefficients. The coefficients of both liquid water and ice particles are integrated over a modified gamma size distribution ($n(L)$ in Equation (10)) with the width equal to 0.1. This is the same size distribution assumed in [22] for Arctic measurements of Atmospheric Emitted Radiance Interferometer (AERI). As for the state-of-the-art, there are no sufficient in situ measurements in Antarctica defining a function for the particle size distribution. Furthermore, as pointed out in [23], the optical properties of ice crystals turn out to be insensitive to the detailed shape of the size distribution if the effective size is defined by Equation (10).

In Table 1 are summarized the instruments, the measurements, and the spectral intervals used by the various algorithms applied in the present study.

**Table 1.** Summary of instruments, measurements, exploited spectral intervals, and parameters derived by the various algorithms used in the presented analysis. SACR, Simultaneous Atmospheric and Clouds Retrieval; CBH, Cloud Bottom Height; CTH, Cloud Top Height; PT, Polar Threshold.

| Instrument | Measure | Exploited Spectral Band | Retrieved Parameters | Algorithm |
|------------|---------|-------------------------|----------------------|-----------|
| REFIR-PAD | spectral radiance | 200–980 cm$^{-1}$ | $D_e$, $OD_v$, $H_2O$/T profiles | SACR |
| REFIR-PAD | spectral radiance | 380–1000 cm$^{-1}$ | phase (ice or liquid) | CIC |
| LiDAR | Backscatt./depolar. signal | 532 nm | CBH, CTH | PT |

The SACR retrieval is initialized by using first guess profiles taken from a seasonal climatology built on the basis of the local radiosonde profiles as described in [10]. The vertical distribution profiles of non-retrieved molecular species, such as $CO_2$, $O_3$, $N_2O$, CO, $CH_4$, and $O_2$, are taken from the IG2 climatology [24]. The backscattering profile used in Equation (12) is obtained by linearly interpolating the closest LiDAR measurements to the timestamp of the REFIR-PAD measurement considered.

The retrieval levels are set at 0 m, 9 m, 67 m, 767 m, 2.767 km, and 4.767 km above the surface for water vapor and at 0 m, 9 m, 67 m, and 767 m for the temperature. This is an optimized choice deriving from previous studies [9,10]. The instrument is separated from the external environment by a 1.5 m chimney. Then, the measurement is sensitive to the temperature of the air in the shelter, which is very different from the external one. For this reason, the first retrieval grid point in the state vector is set up inside the shelter, and the second one is immediately outside the shelter. In the a prioriVCM $S_a$, these two elements are considered as fully uncorrelated. This choice allows fitting the $CO_2$ band of the spectrum better, around the $\nu_2$ central frequency at 667 cm$^{-1}$.

In order to constrain the atmospheric profiles in the SACR inversion, radiosonde profiles are assumed as a priori information. The profiles are linearly interpolated in the time domain to the timestamp of the REFIR-PAD measurement. The a priori error assumed is 50% on water vapor profiles and 1% on temperature profiles. A correlation length of 2 km is used to regularize both profiles above 9 m. For ice clouds, a value of 20 μm is used as a priori for the effective diameter and a value of one for the optical depth, as indicated in previous studies [25,26]. For water clouds, an a priori effective diameter of 10 μm and an optical depth of one are assumed. This choice is made considering that in situ cloud measurements between 75°S and 80°S, over the Ross Ice Shelf and the Ross Sea, revealed that water clouds have a bimodal droplet size distribution with effective diameters in the range 13–17 μm [27]. Furthermore, measurements over coastal Antarctica report values below 20 μm [28]. Nevertheless, as mentioned in [5], clouds show a smaller effective droplet size over land as compared to the ocean. Therefore, it is reasonable to assume that in the middle of the Antarctic Plateau, they may have smaller effective diameters. The a priori error on cloud parameters was set equal to 100% in order to avoid over constraining of the retrieval.

## 4. Results

### 4.1. Validation of the Atmospheric Profiles

Temperature profiles and precipitable water vapor, derived from REFIR-PAD measurements within 1 h from the daily local radiosonde (12:00 UTC), are validated. Figure 4 compares the retrieved temperatures with those obtained from the radiosonde profiles, convolved with the REFIR-PAD Averaging Kernels (AKs) by [14]:

$$\mathbf{AK} = [\mathbf{K}^T\mathbf{S}_y^{-1}\mathbf{K} + \mathbf{S}_a^{-1}]^{-1}\mathbf{K}^T\mathbf{S}_y^{-1}\mathbf{K} \qquad (13)$$

Only the first three levels above the instrument (3342, 3300, and 4000 m) are analyzed. The retrieved temperatures are in good agreement with the radiosonde measurements, as confirmed by the large correlation coefficient (higher than 95%). Figure 5 shows the relative differences in percentage between the Precipitable Water Vapor (PWV) derived from the radiosonde profiles (convolved with the

REFIR-PAD AKs) and that calculated from the retrieved profiles, as a function of CBH. The differences increase for a small CBH, i.e., in the presence of clouds closer to the surface. For the lower clouds, an average negative bias of 29% is found.

For clouds higher than 0.5 km, the negative bias is reduced to ~18%, further decreasing to 15% when considering clouds above 0.8 km. Looking into the seasonality, the winter time bias reaches a value of 28%, significantly larger than the 11% observed during summer time. Therefore, the observed bias cannot be attributed solely to the well-known dry bias affecting radiosonde measurements [29–31]. Indeed, this effect is generally smaller in amplitude and increases with increasing insolation. The observed bias is rather attributed to the supersaturation of ice. As pointed out in [32], supersaturation is not measured by the radiosondes as their sensors' surface acts as a nucleation site upon which vapor condenses. Reference [32] found very good agreement between two standard radiosonde sensors, launched simultaneously towards a cirrus cloud. However, both sensors measured a Relative Humidity (RH) 25–30% smaller than a simultaneous frost point hygrometer, able to measure ice supersaturation inside the cirrus cloud. The average PWV negative bias due to the 30% dry bias found by [32] in the RH measured by the radiosondes turns out to be equal to 30% when clouds are very close to the surface (solid blue line in Figure 5) and equal to 23% when their bottom height is placed at 0.5 km above the surface (dashed blue line in Figure 5). These values agree quite well with the average PWV negative biases found in the data: 29% when PWV integration starts from the surface (green solid line in Figure 5) and 18% when PWV integration starts from 0.5 km above the surface (dashed green line in Figure 5). As shown in [32], the magnitude of the error related to the ice-saturation curve increases as temperature decreases. Thus, the high ice-supersaturation increases the bias. For this reason, the lower temperatures occurring in winter time cause a larger bias as compared to summer time. This is exactly the effect observed in our dataset.

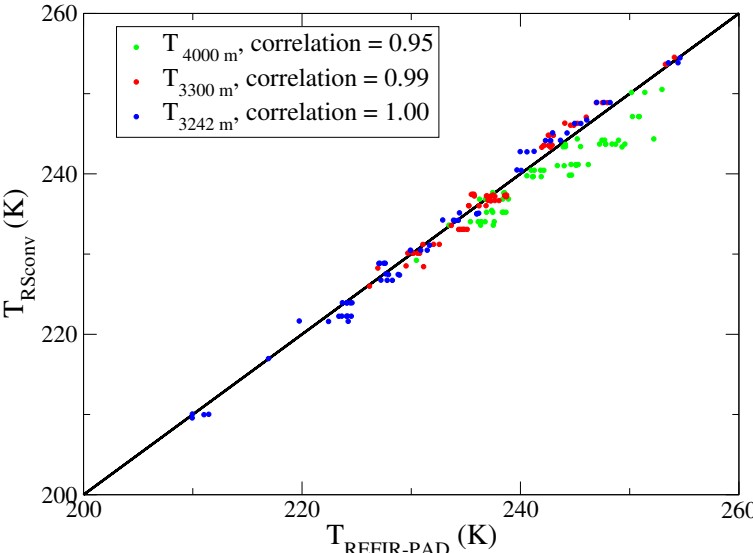

**Figure 4.** Scatter plot of the retrieved temperature versus the temperature measured by radiosondes (convolved with REFIR-PAD averaging kernels). The model first three levels above the instrument are shown (blue, green, and red circles).

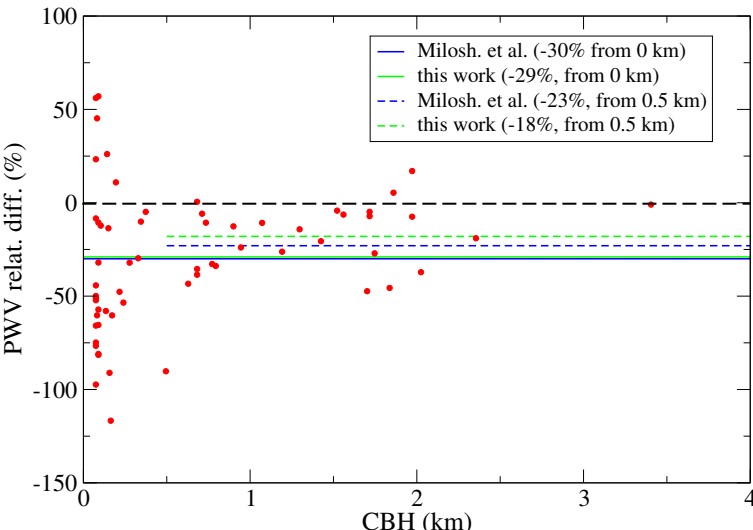

**Figure 5.** Relative differences in percentage, as a function of the CBH, between radiosonde and retrieved Precipitable Water Vapor (PWV) profiles, for the same measurements of Figure 4. The horizontal lines denote the mean bias estimated in this work (green) and the bias that would be produced by a 30% dry bias in the RH measured by radiosonde profiles (blue), as estimated in [32]. Biases are computed for CBH starting from the surface (solid lines) and starting from 0.5 km above the surface (dashed lines).

### 4.2. Case Study of 1 July 2014

Figure 6 shows some parameters retrieved from the REFIR-PAD measurements of 1 July 2014 (Antarctic winter). On this day, the cirrus cloud shown in Figure 2 overpasses the Concordia site. Specifically, Figure 6 shows the retrieved cloud parameters $D_e$ and $OD_v$, the PWV, the average cloud temperature ($T_{cld}$) (calculated as the average of the retrieved temperature profile between CBH and CTH), the temperature at 9 m above the ground ($T_{9m}$), and the temperature of snow at a depth of 5 cm (surface temperature, $T_{surf}$). $T_{surf}$ is extracted from the meteorological database of the Antarctic Meteo-Climatological Observatory. The lowest panel of Figure 6 shows the final value of the reduced $\chi^2$, confirming the good quality of the fit. The effective diameter of the ice crystals ranges between 15 and 40 μm , while $OD_v$ ranges between 0.2 and 1.1.

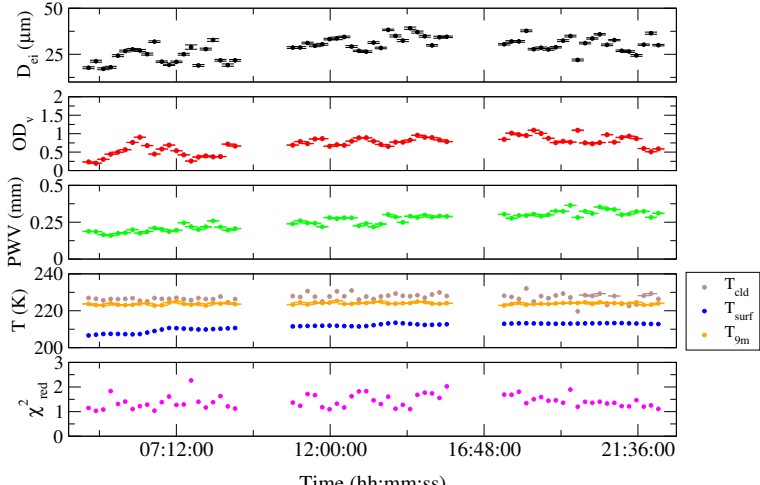

**Figure 6.** From top to bottom: retrieved effective diameter of ice crystals ($D_{ei}$), Optical Depth ($OD_v$), Precipitable Water Vapor (PWV), temperatures of the cloud ($T_{cld}$), of air at 9 m ($T_{9m}$) and of the surface ($T_{surf}$), and reduced $\chi^2$. Data refer to the measurements acquired on 1 July 2014 during the overpass of the ice cloud shown in Figure 2.

PWV is smaller than 0.4 mm, as is typical for the extremely dry conditions of the Antarctic winter. From the fourth panel of Figure 6, the temperature inversion can be appreciated: $T_{surf}$ is always the coldest (206–213 K), whereas the cloud temperature ($T_{cld}$) is warmer (222–232 K) than $T_{9m}$ (220–225 K). This effect is typical for the Antarctic Plateau in winter time. The temperature usually reaches the maximum value at an altitude of about 0.4–0.5 km above the surface. Furthermore, $T_{surf}$ increases from 206 to 212 K as the cirrus optical depth increases, probably because of the cloud warming effect. As an example, Figure 7 depicts the atmospheric conditions around 12:00 UTC of the selected day. The left panel shows the Range Corrected Signal (RCS, $R(z)$), obtained from the LiDAR backscattering signal $S(z)$ as $R(z) = \log(S(z)z^2)$. The signature of a cirrus cloud is clearly visible in $R(z)$, starting from ∼1.9 km above the surface. Indeed, the retrieved CTH and CBH are ∼2.5 km and ∼1.9 km from the surface, respectively (see the dashed horizontal lines in Figure 7). Figure 7 also shows the profiles retrieved from the REFIR-PAD measurement acquired on 1 July 2014 at 12:12 UTC, the closest measurement to the daily radiosonde launch time. The center and the right panels refer to water vapor and temperature profiles, respectively. They show the retrieved (red), the initial guess (green), and the original radiosonde (blue) profiles. The orange lines represent the radiosonde profiles after convolution with the REFIR-PAD averaging kernels. On average, the retrievals provide 3.4 Degrees Of Freedom (DOFs, trace of the AK matrix) for water vapor profiles and 2.8 DOFs for temperature profiles. The retrieved profiles are in very good agreement with the AK-convolved radiosonde profiles, except for the water vapor mixing ratio at 2.767 km. For this specific measurement, the retrieved $OD_v$ of the ice cloud turns out to be (0.678 ± 0.004), and the effective diameter of ice crystals is (34.2 ± 0.2) μm.

The top panel of Figure 8 shows the spectral radiances simulated by SACR at the last retrieval iteration (red) and measured by REFIR-PAD on 01/07/14 at 12:12 UTC (black). The bottom panel shows the differences between observed and simulated radiances (green), as well as the measurement error bounds (black), showing the quality of the fit. Consistently, the final value of the reduced $\chi^2$ is 1.27, i.e., a relatively small value.

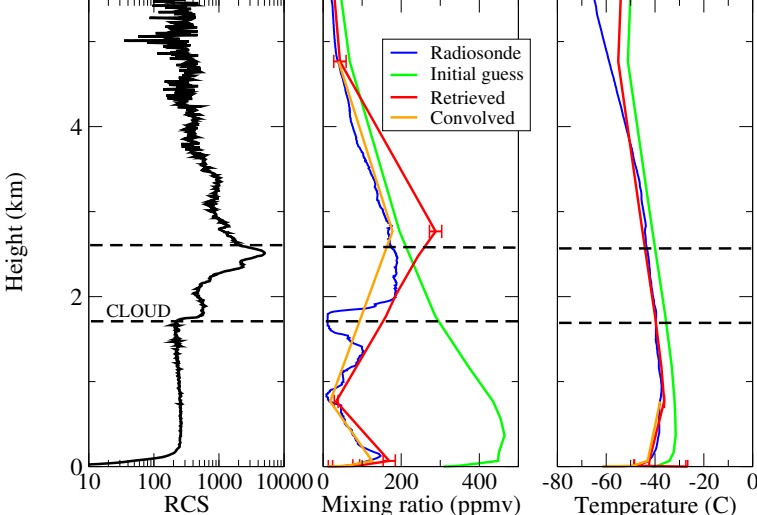

**Figure 7.** **Left panel**: Range Corrected Signal (RCS) of the LiDAR, acquired on 1 July 2014, around 12:00 UTC. The signature of a cirrus cloud is clearly visible starting from ∼1.9 km above the surface. **Middle panel**: water vapor initial guess profile (green), retrieved profile from the REFIR-PAD measurement (red), radiosonde (blue), and Averaging Kernel (AK)-convolved (orange) radiosonde profiles. **Right panel**: as in the middle panel, but for temperature profiles.

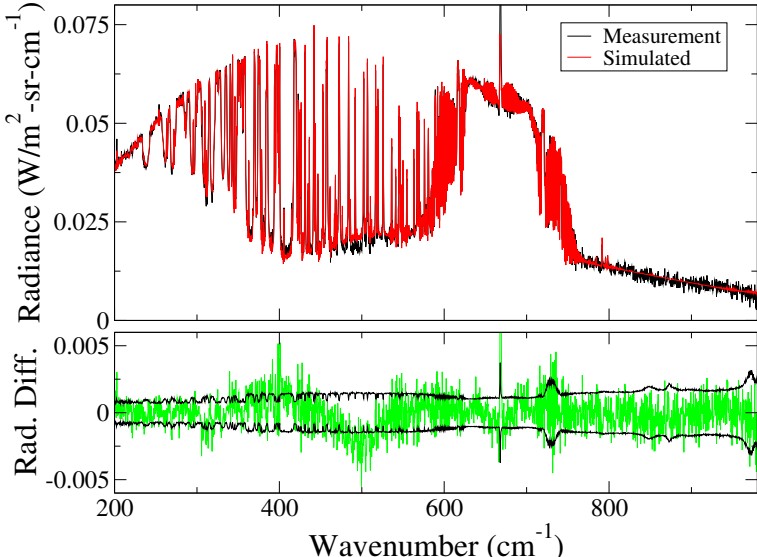

**Figure 8. Top panel**: spectral radiance simulated by SACR (red) and measured by REFIR-PAD on 1 July 2014 at 12:12 UTC (black). **Bottom panel**: difference between observed and simulated radiances (green) and measurement error bounds (black).

*4.3. Results from the Whole Dataset*

4.3.1. Ice Clouds

The ice cloud optical depths of the full dataset range from 0.02 to 4.9. An average $OD_v$ of 0.76 is found, which is in good agreement with the values provided in [25,26]. In autumn, optical depths (average value 1.15) are larger than in summer (average value 0.52). In winter and spring, optical depths take intermediate values, with averages of 0.78 and 0.68, respectively. Figure 9 shows the behavior of the retrieved ice cloud $OD_v$ as a function of other retrieved or computed parameters.

The CBH is plotted in the lower right panel of Figure 9. For cirrus clouds, it mostly ranges from ~70 m to ~3 km above the surface, i.e., very close to the Antarctic tropopause, as also found in [4]. The cloud geometrical thickness ranges from ~10 m to ~6 km. However, it is mostly below 4 km.

The upper right panel of Figure 9 shows the distribution of the Ice Water Path (IWP) calculated from the relationship given by [26]:

$$\mathrm{OD}_{\mathrm{FIR},\nu} = \frac{3 \cdot \mathrm{IWP}}{D_{\mathrm{ei}}\rho_i} \frac{\langle Q_{\mathrm{e}}\rangle_{\mathrm{FIR},\nu}}{2} \tag{14}$$

where $\rho_\mathrm{i} = 917 \text{ kg/m}^3$ is the density of pure ice. In the case of liquid water, IWP is replaced by the Liquid Water Path (LWP), $D_{\mathrm{ei}}$ by the effective diameter of water droplets $D_{\mathrm{ew}}$ and the ice density by the water density $\rho_\mathrm{w} = 1000 \text{ kg/m}^3$. The IWP mostly ranges between 1 and 30 g/m$^2$, as also found in [26], with few outliers showing values down to 0.012 g/m$^2$ and up to 175 g/m$^2$. The $OD_v$ is fitted, as a function of IWP, with a power law of the form $\mathrm{OD}_v = a \cdot \mathrm{IWP}^b$. The fit values for $a$ and $b$ are $(0.211 \pm 0.004)$ and $(0.655 \pm 0.008)$, respectively, when IWP is expressed in g/m$^2$ and the mean relative error on the $OD_v$ is of the order of 20%. A similar approach was used by Heymsfield et al. [33] for mid-latitude and tropical cirri, with coefficients $a$ and $b$ equal to 0.069 and 0.830, respectively. Figure 9 shows that the fit represents the data with good accuracy only for IWP values greater than 1 g/m$^2$.

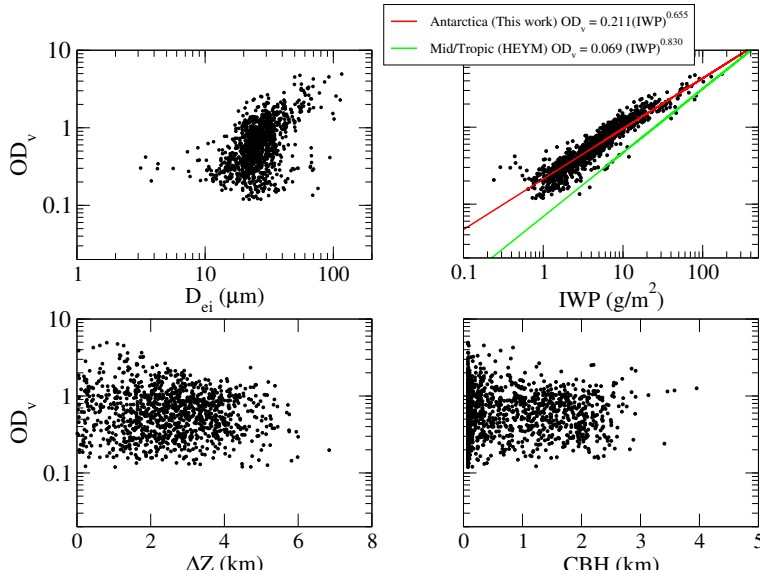

**Figure 9.** Cirrus cloud retrieved Optical Depths ($OD_v$) versus other retrieved or computed parameters: crystals effective diameter $D_{ei}$, Ice Water Path (IWP), cloud thickness $\Delta Z$, and Cloud Bottom Height (CBH). Power laws between $OD_v$ and IWP are shown with solid lines in the upper right panel.

The effective diameter mostly ranges between 10 and 50 μm, with most of the occurrences in the interval 20–30 μm (upper left panel of Figure 9) and an average value of 28 μm. The diameters show a slight seasonality, being smaller during winter and spring (26 and 21 μm) and larger in autumn and summer (34 and 31 μm), when clouds are respectively colder and warmer. The diameters show a moderate correlation with the optical depths of 0.5. Table 2 compares this value with previous results from measurements made in different locations in Antarctica. The current average effective diameter is in very good agreement with the results of Walden et al. (2001) [26]. These authors photographed ice crystals under a microscope at the South Pole and reported an average diameter of 30.4 μm, with a median of 28.2 μm. In [21], the authors used an advance particle imager deployed at the South Pole during early February 2001. This instrument allowed measuring the crystal radius for various crystal habits; thus, the authors did not actually supply an effective crystal radius. For this reason, it is not rigorous to compare directly the results of [21] despite their similarity. In [25], the author collected radiometersonde data (measurements of infrared flux profiles) at the South Pole in the period between 1959 and 1963, suggesting that winter ice clouds have effective diameters in the range 8–32 μm, moderate emissivity of the order of 0.6, and optical depth of the order of one. In [34], the authors suggested that the ice clouds in summer and winter are composed of crystals with effective diameters equivalent to about 24.7 and 11.2 μm. Unfortunately, the study of [35] was based only on a small subset of 26 measurements acquired by REFIR-PAD in 2013 from Concordia. Thus, their dataset is much smaller than that considered in the present work. Furthermore, their retrieval was based only on spectral microwindows, rather than on a broad spectral interval, as we assume in this study. The different setup of the two studies justifies the differences observed in the results. Finally, in [36], surface-based observations were performed of cloud and precipitating particles on the Avery Plateau, on the Antarctic Peninsula, from 25 November to 13 December 1995, finding diameters between 20 and 200 μm.

**Table 2.** Cloud particles' effective diameters at different locations in Antarctica available in the literature.

| Authors | Location | $D_e$ (µm) | Crystals Type |
|---|---|---|---|
| This work | Concordia (2013–2014), 1439 spectra | 28 | Columns |
| | | 55 | Plates |
| | | 51 | Bullet rosettes |
| | | 35 | Aggregates |
| Lawson et al. (2006) | South Pole (February 2001) | 21 ($D_{VA}$) * | Columns |
| | | 38 ($D_{VA}$) * | Plates |
| | | 54 ($D_{VA}$) * | Bullet rosettes |
| Walden et al. (2003) | South Pole (winter 1992) | 20 | Columns |
| | | 30 | Plates |
| | | 50 | Bullet cluster |
| Mahesh et al. (2001) | South Pole (1992) | 30 | |
| Stone (1993) | South Pole (1959–1963) | 8–32 | All crystals |
| Lubin and Harper (1996) | South Pole (summer, 1992) | 25 | |
| | South Pole (winter, 1992) | 11 | |
| Maestri et al. (2019) | Concordia (2013), 26 selected spectra | 50 | Columns |
| | | 40 | Plates |
| | | 38 | Bullet rosettes |
| | | 50 | Aggregates |
| Shimizu (1963) | Byrd Station (winter, 1961) | 100–1000 (length) | Long columns |
| Lachlan-Cope et al. (2001) | Avery Plateau (1995) | 20–200 | All crystals |

* Mean diameter of the equivalent sphere.

### 4.3.2. Water Clouds

Figure 10 shows the correlation between the water cloud $OD_v$ and the other retrieved parameters. $OD_v$ mostly ranges between one and 10, with an average value of four.

The effective diameters of water droplets (upper left panel of Figure 10) are generally smaller than 20 µm, with an average value of 9 µm. However, the results show two cloud distributions with effective diameters below and above 5 µm (green and red dots, respectively).

The top right panel of Figure 10 shows that water clouds are characterized by LWP ranging from 0.05 g/m$^2$ to 145 g/m$^2$, with most of the values between 1 and 40 g/m$^2$. Since LWP is strictly related to $D_{ew}$ (Equation (11)), two separated distributions can be found for diameters smaller and larger than 5 µm (blue and black solid lines). By analogy with the ice cloud analysis, a power law fit is applied to the two distributions obtaining $a = (0.261 \pm 0.014), b = (0.809 \pm 0.023)$ for $D_{ew} \geq 5$ µm and $a = (1.136 \pm 0.036), b = (0.926 \pm 0.018)$ for $D_{ew} < 5$ µm. The calculated relative errors for $OD_v$ give 29% and 13% for the two distributions, respectively.

The CBH (bottom right panel) varies from ∼70 m to ∼1 km above the surface. The cloud thickness (bottom left panel) varies from ∼10 m to 7 km; however, most of the values are above 2 km.

Figure 11 shows the behavior of the retrieved effective diameters of ice crystals (black) and water droplets (red) as a function of the cloud temperature $T_{cld}$ (upper panel) and of the surface temperature $T_{surf}$ (lower panel). The $T_{cld}$ of ice clouds ranges between 220 and 245 K, while water clouds are warmer, ranging between 232 and 250 K. The low $T_{surf}$ below 210 K occur during Antarctic winter, along with the temperature inversion. Because of this effect, the cloud becomes warmer than the surface. As expected [37], the cloud particle effective diameters increase with increasing cloud temperature. Since the warmest atmospheric layers are close to the surface in summer, most of the water clouds in the dataset show up during the Antarctic summer at heights below 2 km.

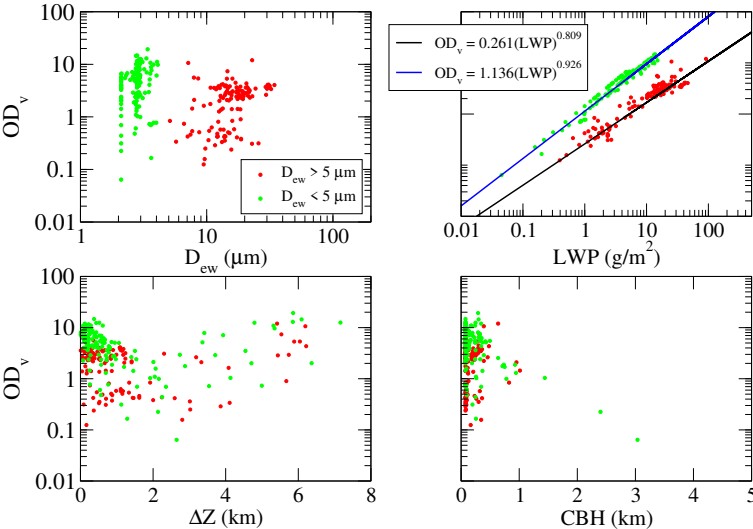

**Figure 10.** As in Figure 9, but for water clouds. LWP denotes the Liquid Water Path and $D_{ew}$ the effective Diameter of water droplets. Green and red dots identifies effective diameters smaller and larger than 5 μm , respectively. The black and blue solid lines on the top right panel indicate power law fittings.

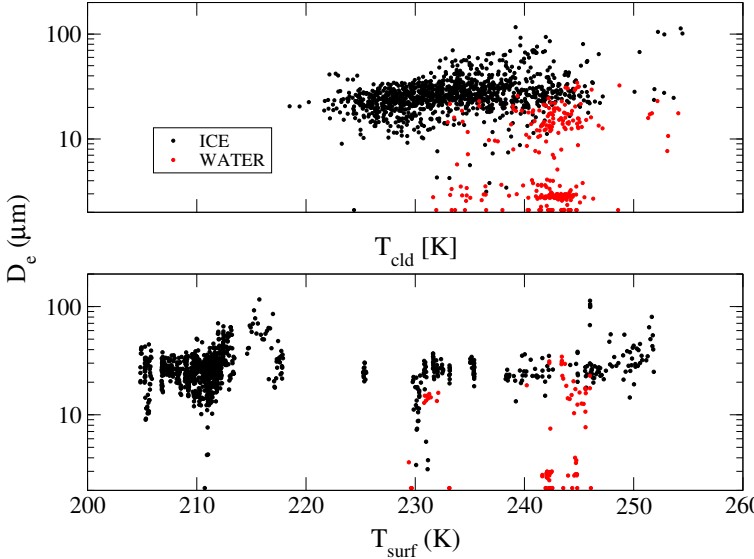

**Figure 11.** Effective diameter of ice crystals (black) and water droplets (red) as a function of the cloud temperature (**top panel**) and surface temperature (**bottom panel**). Data refer to the full analyzed dataset.

*4.4. Sensitivity of Retrieved Parameters to the Crystal Habit*

In ice cloud retrievals, the noise-induced errors are, on average, 1.3% for optical depth and 2.4% for the effective diameter. This error component is, however, smaller than the systematic uncertainty caused by the assumption of a specific crystal habit in the cloud model. To quantify the systematics, the retrieval is repeated by assuming various ice particle habits, as illustrated in [21]. Namely, the following additional crystal habits are considered: complex aggregates, bullet rosettes, and plates. The average retrieved diameters are 35 μm for aggregates, 51 μm for bullet rosettes, and 55 μm for plates, as also shown in Table 2. The upper panel of Figure 12 shows the differences between the retrieved $OD_v$ for solid columns and the other habits. A limited subset of measurements, covering July and December 2014, is shown. The differences in the effective diameters are shown in the lower panel of the same figure. For $OD_v$ smaller than one and $D_{ei}$ smaller than 20 μm , the impact

of the shape assumption is small. The difference observed in the retrieved $OD_v$ is on average $-11\%$, with the largest differences down to $-20\%$ for large $OD_v$. The difference in the retrieved $D_{ei}$ is on average $-50\%$, increasing to $\pm100\%$ for $D_{ei}$ larger than 15 μm.

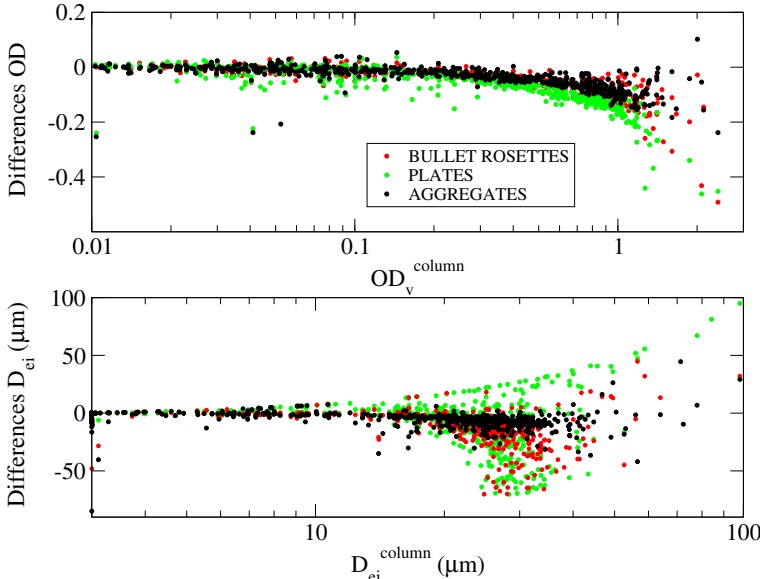

**Figure 12.** Differences in the retrieved optical depth (**top panel**) and in the effective diameters (**bottom panel**), obtained when assuming different ice particle habits with respect to the default solid column habit. Data refer to a subset of measurements, covering July and December 2014.

## 5. Assessment of Cloud Forcing

In the longwave range, the difference between the cloudy and clear sky downwelling fluxes yields the cloud forcing parameter at the surface [38]:

$$C_{LW} = F_{cloud} - F_{clear}. \tag{15}$$

where $F_{cloud}$ is the downwelling flux in the presence of the cloud and $F_{clear}$ is the flux obtained when the cloud is removed from the column [39]. $F_{cloud}$ and $F_{clear}$ are computed by using the Gauss quadrature method. The spectral radiance $S(\nu, \mu)$ is computed at three zenith angles $\mu_i$ (77.740°, 53.805°, and 24.299°), corresponding to the zeroes of the Legendre polynomials. The integration over the solid angle is then obtained as the sum of the radiances, weighted according to the tabulated Gauss coefficients $\delta_i$ [40]:

$$F = 2\pi \int_{\nu_1}^{\nu_2} \sum_{i=1}^{3} S_i(\nu, \mu_i)\delta_i d\nu, \tag{16}$$

where the factor $2\pi$ accounts for the integration on the azimuth angle. The left panel of Figure 13 shows the total downwelling longwave flux at the surface, integrated over the range 100–2500 cm$^{-1}$, as a function of the Water Path (WP; Equation (14)). Black and red dots refer to ice and water clouds, respectively. The right panel of the figure shows the percentage corresponding to the FIR component (100–667 cm$^{-1}$) over the total longwave downwelling flux. The total flux ranges between 70 and 220 W/m$^2$, and the FIR contribution varies from about 56%, for WP around 100 g/m$^2$, up to 75% for WP close to 0.1 g/m$^2$.

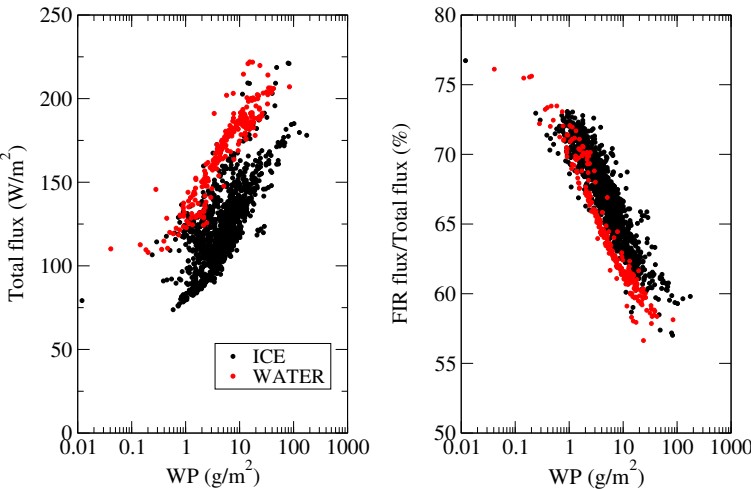

**Figure 13. Left panel**: Total longwave downwelling flux as a function of the Water Path (WP) for ice (black) and water (red) clouds. **Right panel**: percentage ratio between FIR (100–667 cm$^{-1}$) and total longwave downwelling fluxes.

Figure 14 shows the total (100–2500 cm$^{-1}$) downwelling cloud forcing at the surface as a function of WP. Cloud forcing is very sensitive to the optical depth [41] and, thus, to the WP. A power-law relationship between longwave cloud forcing and WP is defined:

$$C_{\text{LW}} = a \cdot \text{WP}^b + c. \tag{17}$$

The best fit provides $a = (81.861 \pm 8.897)$, $b = (0.151 \pm 0.013)$, $c = (-74.752 \pm 9.134)$ for ice clouds and $a = (187.972 \pm 39.732)$, $b = (0.107 \pm 0.019)$, $c = (-161.536 \pm 39.918)$ for water clouds. The mean relative errors for the ice and water cloud forcings are 18% and 9%, respectively.

The parametrization is accurate for IWP and LWP smaller than 100 and 50 g/m$^2$, respectively. The fitting curves are plotted in Figure 14 with solid green and blue lines. The total cloud forcing increases with increasing cloud WP, ranging between 0.2 and 87.3 W/m$^2$ for ice clouds and between 0.9 and 125.7 W/m$^2$ for water clouds. The average forcing by ice clouds is $(31 \pm 7)$ in 2013 and $(29 \pm 6)$ W/m$^2$ in 2014. The forcing by water clouds, only occurring in summer time, is larger than that of ice clouds reaching, on average, $(64 \pm 12)$ W/m$^2$ in 2013 and $(62 \pm 11)$ W/m$^2$ in 2014. The average cloud forcing over the two considered years amounts to $(46 \pm 9)$ W/m$^2$. The fluxes' uncertainty is computed as the standard deviation of the fluxes within hourly time slots. The fluctuations in the hourly sets account for the atmospheric variability, which has a much larger effect on the computed values than the retrieval error.

Table 3 compares the computed cloud forcings with the literature. The average ice cloud forcing is in agreement with the mean forcing of $(38 \pm 3)$ W/m$^2$ found within the Surface Heat Budget of the Arctic Ocean (SHEBA) project (77° N, 165° W) by [39]. Furthermore, the ice cloud forcings for the years 2013 and 2014 are similar to the results from the extended Advanced Very High Resolution Radiometer (AVHRR) polar pathfinder APP-xdataset (1985–93) by [41]. Indeed, the authors found a longwave cloud forcing of about 30 W/m$^2$ at 75°S (the latitude of the Concordia station). This value increases to 35–40 W/m$^2$ in summer time. Further measurements can be considered for comparison. For instance, the values of 24 and 20 W/m$^2$ found by [38] at South Pole (Antarctica, 90°S) in 1992 and 2001, respectively, are very close to this work average values when the contribution of summer water clouds is not accounted for. In the Arctic, at Barrow, Alaska (71.3°N, 156.6°W), Reference [42] found a cloud forcing value of 44 W/m$^2$. Other measurements are also reported at lower latitudes, which give mean forcing values ranging between 25 and 47 W/m$^2$.

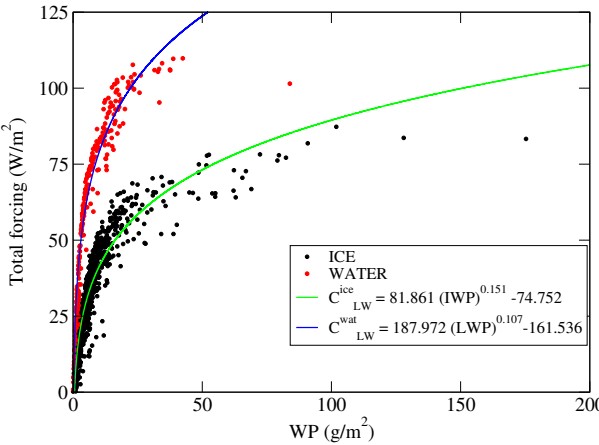

**Figure 14.** Total longwave radiative forcing at the surface for ice (black) and water (red) clouds. Solid lines represent power law fittings of ice cloud data (green solid line) and of water cloud data (blue solid line).

**Table 3.** Cloud forcing at the surface from Polar region campaigns available in the literature. SHEBA, Surface Heat Budget of the Arctic Ocean.

| Authors | Location | Lat, Lon | Alt (m a.s.l.) | Forcing (W/m$^2$) |
|---|---|---|---|---|
| This work | Concordia (2013–2014) | 75°S, 123°E | 3233 | 30 (ice clouds only) |
| This work | Concordia (2013–2014) | 75°S, 123°E | 3233 | 46 (ice + water clouds) |
| Town et al. (2005) | South Pole (2001) | 90°S | 2835 | 20 |
|  | South Pole (1992) | 90°S | 2835 | 24 |
| Pavolonis and Key (2003) | Antarctic cont. (1985–1993) | 75°S |  | ~30 |
| Intrieri et al. (2002) | SHEBA (Nov. 1997–Oct. 1998) | 77°N, 165°W | 0 | 38 |
| Allan (2000) | Barrow, Alaska (1992) | 71.3°N, 156.6°W | 0 | 44 |

## 6. Summary of the Results

The synergy between spectrometer and LiDAR measurements is exploited to characterize the radiative properties of Antarctic ice and water clouds. A backscattering/depolarization LiDAR and a Fourier transform spectroradiometer (the Radiation Explorer in the Far Infrared-Prototype for Applications and Development (REFIR-PAD)) are installed at the Concordia station in Antarctica. Starting from 2012, these instruments have been acquiring colocated measurements in continuous and unattended mode. The acquired data constitute large and unique databases of backscattering profiles and of downwelling spectral radiances in the 100–1500 cm$^{-1}$ (6.7–100 μm) interval, with a spectral resolution of 0.4 cm$^{-1}$.

A data processing scheme is set up to operate automatically on LiDAR and spectroradiometer measurements. First, a Cloud Identification and Classification (CIC) algorithm is applied to the spectral radiance measured by REFIR-PAD. The algorithm determines whether a cloud is present in the field-of-view of the instrument during the measurement. If a cloud is detected, the CIC also discriminates the thermodynamic phase of the cloud (liquid/solid). In parallel, the so-called Polar Threshold (PT) algorithm is applied to the nearest (in time) LiDAR backscattering profile. The PT algorithm determines the cloud top and bottom heights. The spectral radiances affected by a cloud presence are then processed by the Simultaneous Atmospheric and Cloud Retrieval (SACR) model. The SACR algorithm is based on the optimal estimation approach [14]. Assuming the PT derived cloud

bottom and top heights, the SACR code determines the cloud parameters (optical depth and effective diameter of ice crystals and water droplets), atmospheric profiles of water vapor and temperature, and a parameter characterizing the instrument line shape. The a priori estimates of water vapor and temperature profiles are obtained from the daily radiosoundings that are carried out at the Concordia station, while the a priori cloud parameters are derived from previous studies. Specifically, the forward model in the SACR retrieval assumes a column-like ice crystal habit and the single scattering properties for cirrus clouds as provided by specific databases developed by Ping Yang and colleagues [13].

The cloudy measurements acquired during the years 2013 and 2014 are analyzed. As far as the results concerning the retrieval of atmospheric parameter, it is noted that the derived temperature profiles are in good agreement with the local radiosonde measurements. The precipitable water vapor measured by radiosondes shows a dry bias as compared to the REFIR-PAD retrieved profiles. The size of this bias, however, is in good agreement with the value expected on the basis of a previous work [32], considering the ice supersaturation.

The retrieved ice cloud optical depths range from 0.1 to three with an average value of 0.76 in the two years of measurements investigated. The effective diameter of ice crystals ranges from 10 to 50 μm, with an average value of 28 μm and a large number of values within 20–30 μm. The particle effective diameter is moderately correlated (correlation ∼0.5) with the optical depth. The total resulting ice water path is mostly between 1 and 30 g/m$^2$. The cloud bottom height ranges from ∼70 m to ∼3 km above the ground and the cloud geometrical thickness between ∼10 m and ∼4 km. These results are in good agreement with those found in [26] at the South Pole.

Water clouds' optical depth ranges between one and 10, with an average value of four. The effective diameter of water droplets shows a double-peaked distribution, with peaks located above and below 5 μm and an average value of 9 μm. The optical depth and the droplet effective diameter show a small negative inter-correlation of −0.2. The total resulting liquid water path ranges from 1 to 40 g/m$^2$. It is shown that, for Antarctic ice and water clouds, a power law can efficiently be used to represent the variations of optical depth as a function of the ice/liquid water path.

The sensitivity of the retrieved parameters to the assumed ice crystal habit is also studied. Assuming crystal shapes as aggregates, bullet rosettes, and plates, the average retrieved effective diameters are 35 μm, 51 μm , and 55 μm, respectively. The average value of 28 μm is found with our baseline assumption of a column-like crystal habit.

Finally, the longwave downwelling flux and cloud longwave downwelling forcing at the surface, in the band from 100 to 2500 cm$^{-1}$ (4–100 μm), are computed. For the considered dataset, the longwave downwelling flux ranges from 70 to 220 W/m$^2$, while the cloud longwave downwelling forcing ranges between 0.2 and 87.3 W/m$^2$ for ice clouds and between 0.9 and 125.7 W/m$^2$ for water clouds. An average forcing of (31 ± 7) and (29 ± 6) W/m$^2$ is found for ice clouds in the years 2013 and 2014, respectively. Water clouds, identified only in the summer measurements, exert a larger forcing in comparison with ice clouds. This forcing is estimated to be of (64 ± 12) and (62 ± 11) W/m$^2$ in the years 2013 and 2014, respectively. The average total (ice and water) cloud forcing on the two investigated years is (46 ± 9) W/m$^2$. This latter result is in good agreement with the mean annual forcing value of (38 ± 3) W/m$^2$ estimated by Intrieri et al. [39]. Moreover, the mean ice cloud forcing values of 31 and 29 W/m$^2$, found for the years 2013 and 2014 are consistent with the results found by Pavolonis and Keys in [41], on the basis of the APP-x dataset. In that work, the longwave cloud forcing at 75°S (the latitude of Concordia station) is reported to be about 30 W/m$^2$, then increasing to values up to 35 and 40 W/m$^2$ in summer time.

The ice and water cloud forcings constitute an important fraction of the total downwelling fluxes in the Antarctic region, and thus, their definition is extremely important in the characterization of surface energy balance [43]. The cloud occurrence and forcing influences the surface temperature and evaporation, especially in the Antarctic Plateau where the air close to the surface is very dry [44]. For this reason, once again, we want to stress the importance of monitoring both the statistics of Polar clouds' occurrence and water vapor and temperature trends.

The results of this work will also support the development of the ESA's ninth Earth Explorer mission named FORUM (Far-infrared Outgoing Radiance Understanding and Monitoring), which is planned for launch in 2026 (https://www.forum-ee9.eu/). FORUM will provide global measurements of spectrally resolved outgoing thermal radiation of the Earth in the range from 100 to 1600 $cm^{-1}$ (6.25–100 μm in wavelength), i.e., including the far infrared portion of the spectrum that, so far, has never been measured as spectrally resolved from space. These measurements will contribute to improving our understanding of the radiative forcing and feedback processes involved in the climate system [45].

**Author Contributions:** Conceptualization, G.D.N. and L.P.; methodology, G.D.N. and T.M.; software, G.D.N., T.M., W.C., and D.M.; validation, G.D.N., T.M., W.C., D.M., L.P., G.B., and M.D.G.; formal analysis, G.D.N., T.M., W.C., and D.M.; investigation, G.D.N., L.P., and T.M.; resources, G.D.N. and L.P., G.B., and M.D.G.; data curation, L.P., G.B., and M.D.G.; writing, original draft preparation, G.D.N.; writing, review and editing, all; visualization, all; supervision, G.D.N., L.P., and M.R.; project administration, L.P., G.D.N., and G.B.; funding acquisition, L.P., G.D.N., and G.B. All authors read and agreed to the published version of the manuscript.

**Funding:** This research was funded by the Italian PNRA (Programma Nazionale di Ricerche in Antartide).

**Acknowledgments:** This research was supported by the Italian PNRA (Programma Nazionale di Ricerche in Antartide) and the Institut polaire francais Paul Emile Victor (IPEV). More specifically, it was developed as a part of the subprojects PRANA (Proprietà Radiative del vapore Acqueo e delle Nubi in Antartide), COMPASS (Concordia Multi-Process Atmospheric Studies), FIRCLOUDS (Far Infrared Radiative Closure Experiment For Antarctic Clouds), and DOCTOR (Dome-C Tropospheric Observer). Data and information on radio sounding measurements were obtained from the IPEV/PNRA Project "Routine Meteorological Observation at Station Concordia". We thank the mentioned institutions for supplying information about other measurements available at Concordia station.

**Conflicts of Interest:** The authors declare no conflict of interest.

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
