# Peer review of "Characterization of the Far Infrared Properties and Radiative Forcing of Antarctic Ice and Water Clouds Exploiting the Spectrometer-LiDAR Synergy"

_remotesensing, doi:10.3390/rs12213574_

Round 1

Reviewer 1 Report

This is an excellent paper. I will only provide minor comments for improvement to the manuscript during revision process. 

a). Fig. 1, the diagram - it may be a good idea to rotate the diagram clockwise by 90 degree, so the diagram flow will be from top to bottom. In the current form, I somehow got confused with the flow. The arrow lines can be thicker, and possibly in different color.

b). Many figure captions should be revised. Such as, Fig. 3, 4, 5, 6, 7, and 13. You used complete sentences in the 1st lines of the Fig. captions, that should not be the case. For example, in the Fig. 3 caption, the words "are shown" should be removed.

c). The sentence in line 323 and 324 may need to re-phrased a bit. To me, it does not flow very well. May consider to modify this sentence.

d). NASA's MODIS project already produced operational water cloud and ice cloud optical data products for about 20 years. You didn't cite Mike King et al. 2003 paper on MODIS cloud data products. In the MODIS cloud optical property retrieving algorithm. Dr. Ping Yang's ice particle size distributions are also used. You should try to make a comparison between your retrieval results with those from the operational MODIS cloud optical property data products over your site, and make an objective assessment, regardless good or bad, about the MODIS data product. My understand with MODIS cloud optical data product is that retrievals are made for cloud optical depths > 1.0.

Author Response

Point 1: Fig. 1, the diagram - it may be a good idea to rotate the diagram clockwise by 90 degree, so the diagram flow will be from top to bottom. In the current form, I somehow got confused with the flow. The arrow lines can be thicker, and possibly in different color.

Response 1: We thank the reviewer for this suggestion. The Figure has been rotated and the arrows have been made thicker, with different colors.

Point 2: Many figure captions should be revised. Such as, Fig. 3, 4, 5, 6, 7, and 13. You used complete sentences in the 1st lines of the Fig. captions, that should not be the case. For example, in the Fig. 3 caption, the words "are shown" should be removed.

Response 2: All captions have been re-phrased.

Point 3: The sentence in line 323 and 324 may need to re-phrased a bit. To me, it does not flow very well. May consider to modify this sentence.
Response 3: The sentence has been re-phrased as follows: “In Autumn optical depths (average value 1.15) are larger than in Summer (average value 0.52). In Winter and Spring optical depths take intermediate values, with averages of 0.78 and 0.68, respectively”.

Point 4: NASA's MODIS project already produced operational water cloud and ice cloud optical data products for about 20 years. You didn't cite Mike King et al. 2003 paper on MODIS cloud data products. In the MODIS cloud optical property retrieving algorithm. Dr. Ping Yang's ice particle size distributions are also used. You should try to make a comparison between your retrieval results with those from the operational MODIS cloud optical property data products over your site, and make an objective assessment, regardless good or bad, about the MODIS data product. My understand with MODIS cloud optical data product is that retrievals are made for cloud optical depths > 1.0.

Response 4: Although we think this is a very good point and a very interesting aspect to investigate, the amount of work required for a detailed comparison is not compatible with the deadline that was set by MDPI for the revision of the present paper (10 days). Given the interesting suggestion by the reviewer, we have already started to draft a ground-based / satellite data compararison covering a longer time interval of 4 years, in order to catch also inter annual variations.

Reviewer 2 Report

I found this paper very difficult to read and follow. I attribute this to poor English and also as a consequence of the authors mixing up background theory, methodology and results. While I believe this is an excellent paper, the grammer and structure must be corrected before it can be properly reviewed.

Author Response

Point 1: I found this paper very difficult to read and follow. I attribute this to poor English and also as a consequence of the authors mixing up background theory, methodology and results. While I believe this is an excellent paper, the grammer and structure must be corrected before it can be properly reviewed.

Response 1: We have extensively worked on the grammar and phrase construction in order to make the paper more readable and easy to follow. Now, the English language should be sufficiently improved. Some work has been done also to re-organize the presented material. As an example, the sub paragraph “Results from the whole dataset” (in the Results section) is now split into two sub-paragraphs “Ice clouds” and “Water clouds”.

Reviewer 3 Report

Well written paper and timely work. My only suggestion would be to have some discussion on what the implications of the results are on policy.

Author Response

Point 1: Well written paper and timely work. My only suggestion would be to have some discussion on what the implications of the results are on policy.

Response 1: We thank the reviewer for this suggestion. Indeed, we think that a
discussion about the importance of monitoring clouds and water vapour and
temperature trends in Antarctica is needed. The discussion added in the conclusions is
the following: “The ice and water cloud forcings constitute an important fraction of the total downwelling fluxes in Antarctic region and thus their definition is extremely important in the characterization of surface energy balance. The cloud occurrence and forcing influences the surface temperature and water vapor evaporation amount, especially in the Antarctic Plateau where the air close to the surface is very dry. For this reason, once again we want to stress the importance of monitoring both the statistics of Polar clouds occurrence, and water vapour and temperature trends”.
Regarding this, we discussed in the conclusions that the results obtained in this work will support the development of the ESA’s 9 th Earth Explorer mission named FORUM (Far-infrared Outgoing Radiance Understanding and Monitoring) , which it will deal with the monitoring of the Earth's atmosphere providing global measurements of
spectrally-resolved Earth’s outgoing thermal radiation, including the far infrared portion of the spectrum that, so far, has never been measured spectrally resolved from space. These measurements will contribute to improve our understanding of the radiative forcing and feedback processes involved in the climate system.

Round 2

Reviewer 2 Report

The authors have failed to correct the English as requested. In their Abstract alone I made over seventy suggested corrections. I read as far as the end of the introduction before giving up on the review. Sentences are too complex with multiple phrases that should be split into multiple sentences. Personal pronoun (We) is generally unacceptable. Tense should generally be past tense.

My suggestions for the abstract is presented below.
______________________________________________________________

Optical and microphysical cloud properties were retrieved from measurements acquired for 2013 and 2014 at the Concordia base in the Antarctic Plateau. These instruments were a the Fourier transform spectroradiometer REFIR-PAD (Radiation Explorer in Far Infrared - Prototype for Applications and Developments) and a backscattering-depolarization lidar.

To identify the cloudiness and assess cloud thermodynamics, the spectral radiance from REFIR-PAD was analysed using a machine learning algorithm, based on Principal Component Analysis, called Cloud Identification and Classification (CIC). Each cloudy scene identified was paired to the nearest acquired lidar backscattering profile which was processed with the Polar Threshold (PT) algorithm to determine cloud top and bottom heights. The REFIR-PAD spectral radiance was then processed by the Simultaneous Atmospheric and Clouds Retrieval (SACR) code using the cloud identification and heights. The SACR algorithm determines cloud optical depth, ice and water effective particle sizes, and the atmospheric vertical profiles of water vapour and temperature.

The analysis determined an average effective diameter of 28 µm with an optical depth 0.76 for the ice clouds. Water clouds were only detected during the austral Summer and the retrieved properties provided an average droplet diameter of 9 µm and average optical depth equal to 4. The estimated retrieval error on is about one percent for the ice crystal/droplet size and two percent for cloud optical depth. The sensitivity of the retrieved parameters to the assumed crystal shape was also assessed and new parametrizations of the optical depth and the longwave downwelling forcing for Antarctic ice and water clouds as a function of the ice/liquid water path are presented. The longwave downwelling flux computed from the top of the atmosphere to the surface ranges between 70 and 220 W/m2. The estimated cloud longwave forcing at the surface is (31 ± 7) W/m2 for ice clouds and (64 ± 12) for water clouds, in 2013 and 2014, respectively. The total average cloud forcing for the two years investigated was calculated to be (46 ± 9) W/m2.

Author Response

Please see the attachment with the cover letter
